# Recurring Weakness in Rhabdomyolysis Following Pfizer–BioNTech Coronavirus Disease 2019 mRNA Vaccination

**DOI:** 10.3390/vaccines10060935

**Published:** 2022-06-11

**Authors:** Motoya Kimura, Jun-Ichi Niwa, Manabu Doyu

**Affiliations:** Department of Neurology, Aichi Medical University, 1-1 Yazakokarimata, Nagakute 480-1195, Japan; kimura.motoya.476@mail.aichi-med-u.ac.jp (M.K.); douyuu.manabu.049@mail.aichi-med-u.ac.jp (M.D.)

**Keywords:** rhabdomyolysis, COVID-19 vaccination, mRNA vaccine, magnetic resonance imaging, weakness

## Abstract

Rhabdomyolysis is a well-known clinical syndrome of muscle injury. Rhabdomyolysis following coronavirus disease 2019 (COVID-19) vaccination has recently been reported. The patients’ weakness gradually subsided and did not recur. Rhabdomyolysis associated with COVID-19 vaccination has not been assessed by repeated magnetic resonance imaging (MRI) within a short time. We report a rare case of an older woman who developed recurring weakness with rhabdomyolysis after COVID-19 vaccination. A 76-year-old woman presented with myalgia 2 days after receiving a third dose of the COVID-19 vaccine. A physical examination showed weakness of the bilateral iliopsoas muscles. Her creatine kinase concentration was 9816 U/L. MRI showed hyperintensity of multiple limb muscles. She was treated with intravenous normal saline. Her symptoms disappeared within 3 days. However, MRI on day 4 of hospitalization showed exacerbation of the hyperintensity in the left upper limb muscles. On day 5 of hospitalization, weakness of the left supraspinatus and deltoid muscles appeared. MRI on day 8 of hospitalization showed attenuation of the hyperintensity in all muscles. Her weakness and elevated creatine kinase concentration disappeared by day 10. Repeated MRI over a short time may be useful to predict potential weakness and monitor the course of COVID-19 vaccine-induced rhabdomyolysis.

## 1. Introduction

Rhabdomyolysis is a well-known clinical syndrome of muscle injury associated with myoglobinuria, electrolyte abnormalities, and often acute kidney injury [1]. Rhabdomyolysis is caused by the breakdown and necrosis of muscle tissue and the release of intracellular content into the bloodstream. There are multiple and diverse causes of rhabdomyolysis [2], including various types of vaccinations [3,4,5,6]. Interleukin (IL)-6 and IL-1β concentrations were reported to be increased in a patient with exertional rhabdomyolysis [7], and the IL-6 concentration was increased in a patient with rhabdomyolysis and coronavirus disease 2019 (COVID-19) [8].

Since the end of 2019, the COVID-19 pandemic, induced by infection with severe acute respiratory coronavirus 2, has led to millions of deaths worldwide. More than 10 billion vaccine doses have been administered. The currently available and most commonly administered vaccines in Japan are two mRNA vaccines: the Pfizer–BioNTech and Moderna vaccines. Large clinical trials have shown that these COVID-19 vaccines are safe and effective. Common adverse events of these vaccines include mild to moderate reactions at the injection site, fever, fatigue, body aches, and headache [9].

Rhabdomyolysis following Pfizer–BioNTech COVID-19 mRNA vaccination has recently been reported [10,11,12,13]. To the best of our knowledge, the patients’ weakness gradually subsided and did not recur, unlike in the current case. Rhabdomyolysis associated with COVID-19 vaccination as assessed by repeated magnetic resonance imaging (MRI) within a short time has not been previously reported. We measured several cytokines over time in our patient to help to elucidate the pathogenetic mechanism of rhabdomyolysis following COVID-19 mRNA vaccination. Multiple cytokines have not been measured simultaneously in patients with rhabdomyolysis following COVID-19 vaccination, but they have been measured in patients with rhabdomyolysis from other causes [7,8]. We report a rare case of an older woman with muscle weakness that completely resolved once and then developed in other muscles owing to rhabdomyolysis after COVID-19 mRNA vaccination.

## 2. Case Presentation

A 76-year-old Japanese woman with hyperlipidemia visited our emergency room with myalgia 2 days after receiving the third dose of the Pfizer–BioNTech COVID-19 vaccine in her left upper arm. The patient had received the first and second doses of the Pfizer–BioNTech COVID-19 vaccine in her left upper arm 7 and 8 months before receiving the third dose. She developed no adverse effects after the previous COVID-19 vaccinations. She began to experience myalgia of the bilateral upper limbs and left leg 1 day after receiving the vaccine; the myalgia was particularly notable in the left leg. She had no myalgia in the right leg. She could not ambulate independently. Her husband helped to take her to bed. The next day, her limb weakness worsened, and she was transferred to our hospital by ambulance. Her regular medications were suvorexant and atorvastatin. Atorvastatin had been prescribed for 10 years, but the dosage had not changed. She had not taken any sporadic medications prior to admission. Several medical check-ups within a few years were unremarkable. She had no recent history of febrile diseases.

On admission, the patient was conscious; her body temperature was 37.7 °C, her heart rate was 72 beats/minute, her blood pressure was 129/68 mmHg, and her percutaneous oxygen saturation was 100%. A physical examination showed mild bilateral weakness of the iliopsoas muscles (Medical Research Council (MRC) scale 4/4) and no evidence of weakness of the deltoid, biceps, or triceps muscles (MRC scale 5/5); tenderness; edema; or rash. Laboratory findings showed that her white blood cell count was 7400 μL, concentration of creatine kinase (CK) was 9816 U/L, CK-MB was 106 U/L, myoglobin was 1802 ng/mL, aspartate transaminase was 164 U/L, alanine transaminase was 56 U/L, blood urea nitrogen was 13.2 mg/dL, serum creatinine was 0.56 mg/dL, and C-reactive protein was 4.45 mg/dL (Table 1). Urinalysis showed 3+ blood. All other laboratory findings, including electrolyte concentrations and antibodies associated with myositis, were unremarkable. Thyroid function was normal. Real-time reverse-transcriptase polymerase chain reaction for COVID-19 was negative on admission. The IL-6 concentration on the second day of admission had increased. This elevated IL-6 concentration decreased to the normal range and tumor necrosis factor (TNF)-α and IL-1β concentrations remained within the normal range over the disease course (Figure 1). MRI of the limbs showed hyperintensity in the left dominant bilateral triceps brachii, left supraspinatus, deltoid, internal obturator, and gluteal muscles, and subcutaneous fluid retention was observed on short-tau inversion recovery images (Figure 2A–C). The iliopsoas muscles could not be scanned during MRI of the extremities because they were outside of the scan range. An electromyogram (EMG) of the affected muscles indicated no evidence of positive sharp waves or fibrillation at rest and small and partly polyphasic motor unit potentials in voluntary contraction. Transthoracic echocardiography, a pulmonary function test, and whole-body computed tomography imaging were unremarkable.

The patient was admitted and treated with intravenous normal saline for rhabdomyolysis and loxoprofen for myalgia. On day 3 of hospitalization, her weakness and myalgia had completely disappeared. However, the next day, follow-up MRI showed slight exacerbation of the hyperintensity of the left upper limb muscles and attenuation of the hyperintensities of the lower limb muscles (Figure 2D–F). Most affected muscles were enhanced, and only the left supraspinatus muscle had homogeneous enhancement (Figure 3A–C). On day 5 of hospitalization, weakness of the left supraspinatus and deltoid muscles (MRC scale 5/4) developed, with swelling and tenderness of the left supraspinatus. A nerve conduction study (NCS) showed that the compound muscle action potentials (CMAP) of the right and left deltoid muscles were 9.210 and 2.540 mV, respectively. MRI on day 8 of hospitalization showed attenuation of the hyperintensity of all muscles. The weakness gradually subsided and then disappeared on day 10 of hospitalization. Electrolytes and renal function were within normal limits during hospitalization. CK, myoglobin, and liver enzyme concentrations improved over the course of hospitalization, and the CK concentration returned to normal on day 10 of hospitalization. The patient was discharged the next day.

MRI showed the disappearance of the high intensity in the lower limb muscles 2 weeks after discharge and in the upper limb muscles 1 month after discharge. An NCS showed that the CMAP of the deltoid muscle had increased to 7.970 mV in the left deltoid muscle 2 weeks after discharge. An EMG was unremarkable 1 month after discharge.

## 3. Discussion

There are multiple and diverse causes of rhabdomyolysis, which can be classified as acquired or genetic. Acquired rhabdomyolysis is due to trauma and exertion, hypoxic injury, infections, hyperthermia, drugs, and toxins [2]. Influenza [3], recombinant zoster [4], tetanus toxoid [5], and combined tetanus, diphtheria, and acellular pertussis vaccination [6] can cause rhabdomyolysis. Adjuvants are commonly used in medical therapy, including in vaccines. An immunologic adjuvant is defined as any substance that acts to accelerate, prolong, or enhance antigen-specific immune responses [14]. Research has suggested that adjuvants in influenza vaccines might be associated with rhabdomyolysis [15]. Rhabdomyolysis following Pfizer–BioNTech COVID-19 mRNA vaccination has recently been reported [10,11,12,13,16]. In these cases, certain additives might have played a role in the development of rhabdomyolysis following COVID-19 mRNA vaccination. COVID-19 vaccine-induced rhabdomyolysis appeared from 5 h to 2 weeks after receiving the vaccine [11,17]. We experienced a case involving an older woman who developed recurring weakness with rhabdomyolysis following COVID-19 mRNA vaccination. Our patient developed her first episode of weakness 2 days after the vaccination and the second episode of weakness 1 week after the third vaccination.

In previously described patients with COVID-19 vaccine-induced rhabdomyolysis, the CK concentration decreased as symptoms improved, and the weakness did not recur during hospitalization. In our patient, the CK concentration gradually decreased to normal, but weakness of the left supraspinatus and deltoid muscles developed after complete improvement of the bilateral iliopsoas weakness. An NCS showed reduced CMAP in the left deltoid muscle. These findings indicate that the course of the CK concentration did not correspond to the course of the patient’s recurring weakness. The continuous intravenous infusion of normal saline and the smaller muscle volumes of the upper than lower limbs might have masked a slight elevation of the CK concentration after the second episode of weakness. There are several possible causes of our patient’s second episode of weakness. First, her muscle weakness during the second episode might have been caused by the development of neuralgic amyotrophy after the vaccination. However, this is unlikely because of the EMG findings. Second, swelling of the left supraspinatus muscle might have caused entrapment neuropathy. However, this possibility cannot explain the weakness of the deltoid muscle because swelling of the supraspinatus does not affect the axillary nerve. Third, her weakness might have been due to myositis. Antibodies targeting skeletal muscle cells might have been generated, leading to myositis [18]. Although this could cause diffuse hyperintensity of the muscles, as was seen in our case, it was too fast to improve the abnormalities without immunotherapy, and myositis-associated antibodies were negative in our patient. Fourth, her weakness could have been due to rhabdomyolysis caused by immunological responses triggered by the COVID-19 mRNA vaccinations. In this situation, myotoxic cytokines might have been released [18]. The first COVID-19 mRNA vaccination primes the innate immune system to mount a more potent response after the second booster immunization [19]. Our patient developed rhabdomyolysis after receiving her third dose of the vaccine, although she did not develop any adverse effects after the first and second vaccinations. We speculate that the rhabdomyolysis most likely caused her muscle weakness because the clinical courses of both weakness episodes were similar in terms of improvement for a few days without immunotherapy.

We measured multiple cytokines to elucidate the pathogenetic mechanism of rhabdomyolysis following COVID-19 mRNA vaccination, which has been unclear to date. Multiple cytokines have not been measured simultaneously in patients with rhabdomyolysis following COVID-19 vaccination, but they have been measured in patients with rhabdomyolysis of other causes. The IL-6 and IL-1β concentrations were increased in a patient with exertional rhabdomyolysis [7], and the IL-6 concentration was increased in a patient with rhabdomyolysis with COVID-19 [8]. The IL-1β concentration did not increase in our patient. These findings suggest that the mechanism of exertional rhabdomyolysis differs from that of vaccine-associated rhabdomyolysis. A previous report described that the COVID-19 mRNA vaccine elicited the production of Th1 cytokines, including TNF [20]. The TNF-α concentration at the time of the second weakness episode in our patient was within the normal limit. This finding indicates that TNF-α was not produced when the second onset of weakness occurred. We speculate that the regional resistance of muscle against cytotoxic functions of cytokines might vary. This could have delayed the time to develop weakness of the left supraspinatus and deltoid muscles in our patient. Further studies involving cytokine measurements are needed to prevent adverse effects following COVID-19 vaccination and provide specific treatment.

MRI is useful for the assessment of the extent and distribution of rhabdomyolysis. The sensitivity of MRI is superior to that of computed tomography and ultrasonography (100%, 62%, and 42%, respectively). Therefore, MRI is considered invaluable in the early diagnosis of rhabdomyolysis [21]. However, MRI for diagnosis of COVID-19 vaccine-induced rhabdomyolysis has only been performed in a few cases [16,22,23]. Moreover, rhabdomyolysis associated with COVID-19 vaccination has not been assessed by repeated MRI within a short time. In a previous study, the findings of repeated MRI within 2 weeks correlated precisely with the clinical and neurological deficits of patients with rhabdomyolysis [24]. Whether repeated MRI in the short term is useful for patients with rhabdomyolysis has not been reported. In our patient, repeated MRI within a few days showed worsening of the high intensity in the left supraspinatus and deltoid muscles before the development of weakness in these muscles. To our knowledge, whether there is a time gap between the occurrence of MRI changes and physical examination changes has not been reported. Slight weakness is not always detected by the use of the MRC scale, although the sensitivity of the MRC scale in the assessment of rhabdomyolysis-associated weakness is unclear. This means that MRI is more sensitive than physical examination. The discrepancy between the physical examination findings and imaging findings in our case might have been related to the high sensitivity of MRI. Repeated MRI over a short time could be useful for predicting potential weakness and monitoring the course of COVID-19 vaccine-induced rhabdomyolysis. Homogeneous enhancement represents diffuse edema in the affected muscles in the initial stage of rhabdomyolysis [25]. We suggest that homogeneous enhancement of the left supraspinatus muscle represents the delayed development of weakness.

## 4. Conclusions

We predicted recurring weakness by performing repeated MRI and thus continued intravenous infusion of normal saline, helping to avoid the need for a return visit because of recurring weakness. Repeated MRI over a short time could be useful for predicting potential weakness and monitoring the course of COVID-19 vaccine-induced rhabdomyolysis. In patients with COVID-19 vaccine-induced rhabdomyolysis, physicians should pay attention to the possibility of recurring weakness when the MRI findings worsen, even if the CK concentration continues to decrease and the weakness initially disappears.

## Figures and Tables

**Figure 1 vaccines-10-00935-f001:**
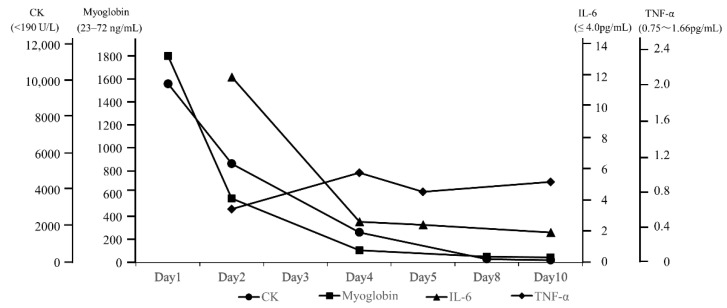
Course of laboratory findings.

**Figure 2 vaccines-10-00935-f002:**
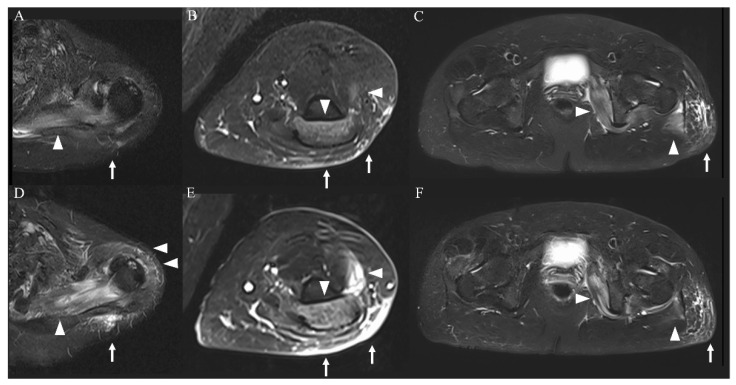
MRI of the extremities (short-tau inversion recovery images). Imaging on admission showed a high-intensity area in the left dominant bilateral triceps brachii, left supraspinatus, deltoid, internal obturator, and gluteal muscles ((**A**–**C**), arrowheads) and subcutaneous tissue ((**A**–**C**), arrows). These findings were more severe in the upper limb muscles ((**D**,**E**), arrowheads and arrows) and were attenuated in the lower limb muscles ((**F**), arrowheads and arrows) on day 4 of hospitalization.

**Figure 3 vaccines-10-00935-f003:**
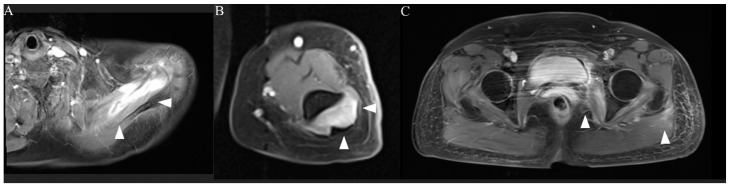
MRI of the extremities (post-contrast T1 images). Only the left supraspinatus muscle was homogeneously enhanced ((**A**), arrowheads). Other affected muscles showed enhancement centered on the rim ((**B**,**C**), arrowheads).

**Table 1 vaccines-10-00935-t001:** Course of laboratory findings and MRC scale.

Parameter	Day 1	Day 2	Day 3	Day 4	Day 5	Day 8	Day 10
CK (<190 U/L)	9816	5416	-	1630	-	164	101
Myoglobin (23–72 ng/mL)	1802	559	-	106	-	52	44
IL-6 (≤4.0 pg/mL)	-	11.9	-	2.6	2.4	-	1.9
IL-1β (≤ 10 pg/mL)	-	≤10	-	≤10	≤10	-	≤10
TNF-α (0.75–1.66 pg/mL)	-	0.59	-	0.99	0.78	-	0.89
MRC	iliopsoas 4/4	iliopsoas 5–/4	all 5/5	all 5/5	supraspinatus and deltoid muscle 5/4	supraspinatus and deltoid muscle 5/5–	all 5/5

## Data Availability

The authors declare that the data supporting the findings of this study are available within the paper.

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
