# Peer review of "Recurring Weakness in Rhabdomyolysis Following Pfizer–BioNTech Coronavirus Disease 2019 mRNA Vaccination"

_vaccines, 2022, doi:10.3390/vaccines10060935_

Round 1
Reviewer 1 Report
case presentation:
for the differential diagnosis of rhabdomyolysis It is important to specify:
- the COVID-19 status of the patient upon admission (i.e. PCR or Antigen test results)
- the patients' regular medications (if any), or sporadic medication uptake prior to admission
- any recent febrile disease
description of findings on physical examination and imaging:
the authors describe weakness of the iliopsoas muscles but there is no MRI image of these muscels. also there are images of triceps and biceps muscles but there is no description of the findings on physical examination.
it could be useful to complete the missing information . if there is discrepance between findings on physical examination and findings on imaging it is intresting to learn which of the two investigations is more sensitive? is there any time gap between these investigations?
Reviewer 2 Report
This is a fairly interesting case report. However, the have been cases of rhabdomyolysis after the case COVID-19 mRNA vaccination documented, so the novelty is missing.
The major problem I see, is that the case description does not support the conclusions. Despite multiple speculations, it is unclear why the course of the CK concentration did not correspond to that of her recurring weakness and the MRI finding. It is also unclear what is the benefit of performing multiple repeated MRI imagining or if there is any at all. How did it affect the treatment and the outcome in this patients?
Reviewer 3 Report
This case report describes a case of rhabdomyolysis following the administration of the third dose of Pfizer vaccine
This is not new: s there are many reported cases of rhabdomyolysis with different types of vaccines and some of them have occurred in second and third doses. There are also reports involving confunding factors such as chronic statine use(United States Department of Health and Human Services (DHHS), Public Health Service (PHS), Centers for Disease Control (CDC) / Food and Drug Administration (FDA), Vaccine Adverse Event Reporting System (VAERS) 1990 - 4/16/2021, CDC WONDER On-line Database. [Apr;2021 ];https://wonder.cdc.gov/vaers.html 2021 4:20–24)
- The strength of this case report is the radiological documentation, with sequential MRI.But this leads the authors to recommend to the authors to perform repeated MRI scans to confirm the evolution of the picture. This recommendation seems to be excessive, especially if there is normalisation of analytical parameters and improvement of symptoms.
- no mention is made of the possibility that vaccine adjuvants may have played a role in rhabdomyolysis, when this is a crucial and established factor in other vaccines.
- what does cytokine seriation add to the case? How useful do the authors think it could be in the management of these patients?
Round 2
Reviewer 2 Report
All my concerns have been addressed as much as it was possible.
Reviewer 3 Report
Thanks
All comments have been already corrected